# Determination of Urinary Pterins by Capillary Electrophoresis Coupled with LED-Induced Fluorescence Detector

**DOI:** 10.3390/molecules24061166

**Published:** 2019-03-24

**Authors:** Wojciech Grochocki, Magdalena Buszewska-Forajta, Szymon Macioszek, Michał J. Markuszewski

**Affiliations:** Department of Biopharmacy and Pharmacodynamic, Medical University of Gdansk, Gdansk 80-416, Poland; grochoo@gumed.edu.pl (W.G.); magdalena.buszewska-forajta@gumed.edu.pl (M.B.-F.); szymon.macioszek@gumed.edu.pl (S.M.)

**Keywords:** capillary electrophoresis, LED induced fluorescence, pterins, biomarkers

## Abstract

Urinary pterins have been found as potential biomarkers in many pathophysiological conditions including inflammation, viral infections, and cancer. However, pterins determination in biological samples is difficult due to their degradation under exposure to air, light, and heat. Besides, they occur at shallow concentration levels, and thus, standard UV detectors cannot be used without additional sample preconcentration. On the other hand, ultra-sensitive laser-induced fluorescence (LIF) detection can be used since pterins exhibit native fluorescence. The main factor that limits an everyday use of LIF detectors is its high price. Here, an alternative detector, i.e., light-emitted diode induced fluorescence (LEDIF) detector, was evaluated for the determination of pterins in urine samples after capillary electrophoresis (CE) separation. An optimized method was validated in terms of linearity range, limit of detection (LOD), limit of quantification (LOQ), intra- and interday precision and accuracy, sample stability in the autosampler, and sample stability during the freezing/thawing cycle. The obtained LOD (0.1 µM) and LOQ (0.3 µM) values were three-order of magnitude lower compared to UV detector, and two orders of magnitude higher compared to previously reported house-built LIF detector. The applicability of the validated method was demonstrated in the analysis of urine samples from healthy individuals and cancer patients.

## 1. Introduction

Pterin and its derivatives are heterocyclic compounds based on the pteridine ring system [1,2,3,4]. They occur naturally in almost all organisms in which they play many important roles such as coenzymes [5], inhibitors [6], sensitizers [7,8], sensors [9], pigments [10,11,12,13,14], and toxins [15]. It has been found that activation of the human immune system during some pathological processes (e.g., cancer, viral infection, renal dysfunction) leads to significantly elevated levels of pterins excreted to urine. Therefore, measurement of urinary pterins concentration might be a promising approach for fast screening and/or early diagnosis of some severe diseases [3,4,16,17,18,19]. The chemical structures of selected pterins are depicted in Figure 1.

Nevertheless, analysis of pterins is very challenging, mainly due to their physicochemical properties. Firstly, a solubility of pterins in water, as well as in various organic solvents, is minimal. As weak basic compounds, pterins are slightly soluble in alkaline solution at relatively high pH (>9). Secondly, their stability is significantly affected by light and temperature [20,21,22]. Therefore, samples have to be stored in non-UV transparent containers at low temperature (preferably below −20 °C) to avoid analytes degradation.

On the other hand, providing such conditions during the analysis might be difficult, and thus, the exposure of samples to light and higher temperature should be maximally reduced. Thirdly, pterins exist in three different oxidation forms, i.e., fully reduced, partially reduced, and fully oxidized (e.g., tetrahydropterin, dihydropterin, and pterin). For quantification of pterins total amount, the sample needs to be either completely reduced or oxidized. Both approaches have been investigated, and it has been found that an analysis of fully oxidized forms is preferable due to their better stability [3]. Finally, pterins are present in biological samples at very low concentration levels that requires using more sensitive detection techniques.

Most of the methods used for determination of pterins in biological samples were based on chromatographic techniques that included paper chromatography [13], reverse-phase [12,23,24], ion-pair [25], hydrophilic interaction [24,26,27], ion-exchange [28], and chiral high-performance liquid chromatography (HPLC) [29]. The second most employed technique in pterins studies was capillary electrophoresis (CE) which has been used for analysis of urine samples [16,18,30,31] as well as insect extracts from *Heteroptera* suborder [32,33]. Standard UV detection has been found suitable for analysis of insect extracts but not for urine samples where pterins concentration is in micro- to the nanomolar range [17]. To overcome this issue, Ma’s group built their own ultra-sensitive laser-induced fluorescence (LIF) detector for the analysis of urine samples since pterins exhibit native fluorescence with an excitation wavelength (λexc) of 325–370 nm and emission wavelength (λem) of 400–460 nm [16,18,31]. The obtained LOD values were as low as 2.5 × 10^−10^ M. On the other hand, there have not been any reported studies using commercially available LIF detectors in CE analysis of pterins. The possible explanation might be the very high price of such instruments which restricts their common usage [34].

This work aimed to evaluate the utilization of commercially available light-emitted diode induced fluorescence (LEDIF) detector coupled with CE for the determination of pterins in human urine samples. So far, all CE approaches in the analysis of urinary pterins have used house-built LIF detectors. Although there are commercially available LIF detectors, their price is significantly higher than LEDIF ones. Therefore, there is a need to find alternative tools for the broader scientific community that is not familiar with manufacturing their instruments or cannot afford expensive analytical equipment. Here, a previously published method was used with minor optimization. Method validation in terms of linearity range, limit of detection, limit of quantification, intra- and interday precision and accuracy, sample stability in the autosampler, and sample stability during freezing/thawing cycle was conducted. Afterward, the validated CE-LEDIF method was successfully applied for the analysis of urine samples from healthy individuals and cancer patients.

## 2. Results and Discussion 

### 2.1. CE Conditions Optimization

Initial CE conditions were adapted from [31] where BGE was an aqueous solution of 100 mM boric acid, 100 mM TRIS, and 2 mM EDTA sodium salt at pH 9.63. A sample consisted of a mixture of 8 pterins after oxidation procedure and was introduced hydrodynamically for 6 s at 50 mbar into the fused-silica capillary (44 cm E.L. and 65 cm T.L.). The separation voltage of 25 kV was applied at positive polarity. Under experimental conditions, a baseline separation was obtained in less than 23 min for all tested analytes. However, due to analytes instability, the total time of pterins analysis should be as short as possible. This is particularly important when the developed method is used for analysis of a considerable number of samples. The influence of using shorter capillary (35 cm E.L. and 56 T.L.), separation voltage (15–25 kV) and temperature of the capillary (20–25 °C) was studied to shorten the time of a single run. Baseline separation was achieved when 17.5 kV was applied throughout the shorter capillary at 25 °C in less than 18 min. Finally, to maximize the sensitivity, the sample injection time was optimized. The injection pressure was kept constant at 50 mbar, and the sample was introduced for 3, 6, and 10 s. Sharp peaks were observed for 3 s and 6 s whereas 10 s sample injection produced broad peaks that indicated capillary overloading. Therefore, 6 s injection time was selected since it gave the highest peaks. 

A representative electropherogram obtained from analysis of pterins standards under optimized conditions is presented in Figure 2. The optimized parameters used for further studies were as follows: 100 mM boric acid, 100 mM TRIS, and 2 mM EDTA sodium salt at pH 9.6 as BGE, fused-silica capillary (50 µm I.D., 35 cm E.L., and 56 cm T.L.), separation voltage 17.5 kV, temperature 25 °C, and hydrodynamic injection at 50 mbar for 6 s. 

### 2.2. Method Validation

The optimized method was then validated in terms of linearity range, LOD, LOQ, intraday and interday repeatability, and accuracy. In this study, dilutions were prepared from the working solution, and an oxidation procedure was performed individually for each concentration level. 

The obtained results are summarized in Table 1. 

The optimized method was linear (R^2^ ≥ 0.996) in a range from 0.3–15 µM while LOD and LOQ values were 0.1 and 0.3 µM, respectively, for all tested analytes. These values correspond to the concentration of the analytes in the sample before the oxidation procedure that led to a further dilution of the sample around 5 times (e.g., the final concentration of the 5 µM sample after oxidation was 1.11 µM). The obtained LOD values were about 400 times higher compared to previously reported results using CE with gravimetric injection and house-built LIF detector [16]. There are several possible explanations of such difference in sensitivity between those two systems. Firstly, it has been demonstrated that λ_exc_ of 325 nm induces the most intense fluorescence for many pterins [35]. Therefore, Ma’s group used a helium-cadmium laser at 325 nm in their house-built LIF detector. On the other hand, the LEDIF detector was equipped with LED at 365 nm. The obtained UV absorbance spectra (see Appendix A) indicates that the optimum λ_exc_ is found at around 350 nm for most of the analytes, which is also consisted with previously published data [36,37]. Secondly, according to the information provided by the manufacturer of the LEDIF detector, its sensitivity in most of the cases is two to five times lower compared to LIF detector when the same λ_exc_ is used [38]. Another study, using house-built laser-based and LED-based systems, has shown one to two orders of magnitude lower sensitivity of LED-based system [39]. In both cases this was a consequence of wider width of wavelengths as well as non-coherent light emitted by the LED. Finally, the emission signals were collected using bond-pass filters in both systems. However, the wavelength range of collected fluorescence was 370–760 nm and 400–539 nm in LED-based system and laser-based system, respectively. Nevertheless, an indication of one main reason of different LODs values obtained by LIF and LEDIF detectors is difficult since both instruments have distinct designs. It should be noticed that no commercially available LIF detector has been used for pterins determination. We believe that utilization of the detectors in which the only variables are the type of the light source or λ_exc_ value (e.g., Zetalif LIF 325 nm, Zetalif LIF 355 nm, and Zetalif LED 365 nm from Picometrics), could explain the effect of these variables on the sensitivity in determination of pterins. 

The precision and accuracy of the developed method were evaluated at three concentration levels, i.e., low, medium, and high-quality control (LQC, MQC, and HQC) and the obtained results are presented in Table 2. 

The intra-day (*n* = 5) and interday (*n* = 15) precision (expressed as %RSD) were acceptable for most of the analytes, and their values varied from 0.8–4.95 and from 0.98–10.57, respectively. However, the results obtained for isoxantopterin did not meet the criteria for analytical standards since the intraday %RSD was as high as 30.45. Afterward, the accuracy (*n* = 5) was calculated based on the linear regression equation. Good results, i.e., 91.70–110.60 % were achieved for all analytes except isoxantopterin (71.20–84.10%). Poor precision and accuracy of the isoxantopterin determination indicates that this analyte is not stable or its measurement can be easily affected by small changes in experimental conditions such as temperature and/or humidity variations. To address this issue a number of stability studies were conducted. 

### 2.3. Stability Study

In the routine analysis, all samples are usually prepared at the same time and then placed in an autosampler where they are waiting to be analyzed. Depending on the time of a single run as well as the total number of the samples, the waiting time might vary from a couple of minutes to several hours. Sometimes, however, samples cannot be analyzed straight away after collection and need to be stored in the freezer. It has been shown that thawing and refreezing of biological samples might potentially change their composition [40,41,42,43,44]. Therefore, it is necessary to determine the stability of the samples, both during storage as well as during lengthy analysis, especially when dealing with very labile analytes like pterins. 

Pterins stability in the autosampler was studied by analyzing the same sample at three-time points, i.e., immediately after preparation, after three, and six hours. The values obtained at the first time point were considered as 100% and were used as a reference to the values at the second and third time point. Also, to evaluate the effect of concentration of the analytes, the experiments were performed at LQC and HQC concentration levels in three replicates for each time point. The results are summarized in Table 3.

Overall, the degradation was negligible for all analytes at a higher concentration level (HQC) after three hours in the autosampler. On the other hand, samples at a lower concentration level (LQC) were more susceptible to degradation where the concentrations of 7-biopterin, 6,7-dimethylopterin, and 7-hydroxylumazin decreased by about 10%. A similar trend was observed after six hours when the composition of the HQC samples did not change significantly while concentration of the analytes in LQC samples dropped slightly. An exception was 7-hydroxylumazin, of which concentration decreased after six hours to 73% and 84% of the initial LQC and HQC level, respectively. 

Afterward, the effect of the freezing/thawing cycle on sample composition was studied by preparing mixtures of pterins at LQC and HQC concentration levels. A part of these samples was collected, oxidized and analyzed whereas, the remaining solutions were frozen. After 24 h, the samples were thawed, and part of them was examined after the oxidation procedure. Finally, the same protocol was repeated after 48 h. From the results presented in Table 4 it can be noticed that the amount of the analytes did not change significantly after the first freezing/thawing cycle whereas, a substantial decrease of several analytes concentration, i.e., pterin, lumazine, isoxantopterin, and 7-hydroxylumazin was observed after the second cycle. This indicates that to provide accurate results samples should not be frozen and thawed multiple times. On the other hand, the poor inter-day repeatability of isoxantopterin cannot be explained based on the stability tests conducted. Therefore, further stability studies, including longer period of time, the influence of temperature and humidity, are required to address this issue. 

### 2.4. Analysis of Urine Samples

Analysis of urine can be challenging since the volume of urine produced varies significantly, depends on the time of day and many other factors such as hydration, age, diet, and physiological conditions of the body. Therefore, the concentration range of metabolites excreted in urine might be relatively wide. 

To evaluate the applicability of the validated method, the analysis of urine samples collected from healthy individuals (*n* = 5) and cancer patients (*n* = 5) was conducted, and the representative electropherograms are presented in Figure 3A,B, respectively. 

In both cases, most of the pterins were quantified (see Table 5). However, there were also some analytes that were not detected. That indicates that those analytes were either present at the concentration below obtained LOD or the samples tested did not contain them. Therefore, the developed method should be cross-validated against more sensitive and selective detection techniques such as mass spectrometry. We also believe that further sensitivity improvement, comparable to this obtained by house-built LIF detector, might be achieved by using online sample preconcentration techniques and thus, LEDIF detection can become a cheaper alternative to LIF detectors.

## 3. Materials and Methods 

### 3.1. Chemicals and Reagents

All pterins standards i.e., 6-hydroxymethylpterin, 6,7-dimethylpterin, xanthopterin, isoxantopterin, pterin, 7-biopterin, D-neopterin, lumazine, and 7-hydroxylumazine were purchased from Schricks Laboratories (Jona, Switzerland). Sodium hydroxide (NaOH), phosphoric acid (H_3_PO_4_), sodium phosphate monobasic (NaH_2_PO_4_), sodium phosphate dibasic (Na_2_HPO_4_), boric acid (H_4_B_2_O_7_), ethylenediaminetetraacetic acid (EDTA) sodium salt, tris(hydroxymethyl)aminomethane (TRIS), potassium iodine (KI), and iodide (I_2_) were obtained from POCH (Gliwice, Poland). Methanol (MeOH) and acetonitrile (ACN) were HPLC grade and purchased from Sigma-Aldrich (St. Louise, MO, USA). Ultrapure water was obtained from Millipore Milli-Q system water system (Bedford, MA, USA). Stock solutions of 1 M NaOH, 1 M H_3_PO_4_, 1 M NaH_2_PO_4,_ 1 M Na_2_HPO_4_, 1 M H_4_B_2_O_7_, 50 mM EDTA sodium salt, 1 M TRIS were prepared in purified water. Working solutions and buffers were prepared by appropriate dilution of stock solution in water and/or organic solvents. 

### 3.2. Instrumentation

All CE experiments were carried out using the 7100 CE system from Agilent Technologies (Santa Clara, CA, USA) coupled with Zetalif LEDIF detector (λ_ex_/λ_em_ = 365/450 nm) from Picometrics (Toulouse, France). Autosampler was thermostated 10 °C below ambient temperature (~8 °C) by using external water bath. Separation of analytes was performed in a fused-silica capillary of 50 µm I.D. and 365 µm O.D. obtained from Polymicro Technologies (Phoenix, AZ, USA). The total length of the capillary was varied, but the detection window for the LEDIF detector was always burnt out 19.5 cm from the outlet end of the capillary. Data acquisition was performed using licensed ChemStation software from Agilent Technologies.

### 3.3. General CE Procedure

A new capillary was conditioned before use by flushing (1 bar) with 1 M NaOH for 15 min, water for 10 min, and BGE for 10 min. On the beginning of the working day, the capillary was rinsed with 0.1 M NaOH, water, and BGE for 10 min, 5 min, and 15 min, respectively. Before each run, the capillary was flushed for 1 min, 1 min, and 5 min, correspondingly. The sample was injected hydrodynamically at 25 mbar. Separation voltage was applied at a positive polarity with the cathode at the detector side. 

### 3.4. Sample Buffer, BGE, and Oxidizing Solution

The BGE and sample buffer’s composition was adopted from [31]. Briefly, sample buffer was prepared by dilution of NaH_2_PO_4_ stock solution in purified water to final concentration of 50 mM and pH was adjusted with phosphoric acid to 7.70. The BGE consisted of a mixture of TRIS (50–200 mM), H_4_B_2_O_7_ (50–200 mM), and 2 mM EDTA sodium salt in water or water/organic solvent mixture. It was prepared freshly every day by appropriate dilution of stock solutions and its pH was adjusted with NaOH. The oxidizing solution of I_2_ in KI was prepared by dissolving I_2_ in the concentrated solution of KI followed by dilution with purified water to obtain the final concentration of 2% I_2_ and 4% KI. To avoid photodegradation, the solution was kept in an amber glass flask wrapped with aluminum foil and stored in the dark. 

### 3.5. Pterins Stock Solutions

Each pterin stock solution was prepared by dissolving 2 mg of standard in 0.3 mL of 1 M NaOH mixed with 9.7 mL of sample buffer. Afterwards, a working solution was prepared by mixing all pterins’ standard stock solutions and diluting them in sample buffer to obtain a final concentration level of 5 × 10^−5^ M each. The working solution was then used for CE method optimization and validation. 

### 3.6. Urine Samples

The urine samples were collected in the Department of Urology, Medical University of Gdańsk, from healthy volunteers with no records of cancer and patients diagnosed with urinary tract cancer. After collection, samples were centrifuged, the supernatant was transferred to fresh plastic tubes that were then kept in a freezer (−80 °C) until they were analyzed. 

All subjects gave their informed consent for inclusion before they participated in the study. The study was conducted following the Declaration of Helsinki, and the protocol was approved by the Bioethics Committee of the Medical University of Gdansk (number of consent: NKBBN/49/2013).

### 3.7. Oxidation Procedure

The oxidation procedure was applied for both pterin standards solution and urine samples. An aliquot of 1000 µL of pterin solution or urine was placed in an amber glass sample vial and mixed with 400 µL of the oxidizing solution and 100 µL of 2 M NaOH. The mixture was then incubated in 4 °C for 30 min in the dark. After the incubation, 500 µL was transferred to a fresh amber glass sample vial and diluted with 500 µL of sample buffer. The sample was mixed and injected directly into the CE.

### 3.8. LOD, LOQ, and Linear Range Determination

The LOD and LOQ values were determined by preparing a serial dilution of the stock solution of the analytes (i.e., 5 × 10^−5^ M, 5 × 10^−6^ M, 5 × 10^−7^ M, and 5 × 10^−8^ M). The concentration of each analyte was the same. Analysis of such prepared dilutions was continued until no analytes were detected. Afterwards, the concentration range between the last dilution where analytes were detected and the dilution where analytes were not detected i.e., between 5 × 10^−7^ M and 5 × 10^−8^ M was studied in detail. The tested concentration levels were 5 × 10^−7^ M, 4 × 10^−7^ M, 3 × 10^−7^ M, 2 × 10^−7^ M, 1 × 10^−7^ M, and 9 × 10^−8^ M. The LOD and LOQ values were determined when signal-to-noise ratio was 3 and 10, respectively.

To determine the linear range, ten different concentration levels were used in three replicates each. The lowest concentration was LOQ while the highest one was the maximum concentration that provided linearity with r^2^ > 0.995. 

## 4. Conclusions

A method’s validation for pterins determination in urine samples using an entirely commercial CE-LEDIF system was presented for the first time. Unlike previously published studies that used house-built LIF detectors, this work did not require any modifications of commercially available instrumentation. Therefore, the proposed approach can be used by the broader scientific community that is not familiar with the engineering/manufacturing of house-built equipment. The sensitivity of CE-LEDIF was lower at about two orders of magnitude compare to the CE-LIF technique, and it was a consequence of employing a less powerful light source as well as a mismatch of the wavelengths used. Despite this drawback, the CE-LEDIF method could be successfully applied for the analysis of real urine samples from cancer patients and healthy subjects. We believe that further sensitivity improvement can be achieved by using online sample preconcentration techniques that might reduce the number of undetected pterins and provide a powerful tool for metabolomic studies. Last but not least, the developed CE-LEDIF method is an excellent example of the ‘green analytical’ approach that helps to minimize the harmful impact of analytical chemistry on the environment.

## Figures and Tables

**Figure 1 molecules-24-01166-f001:**
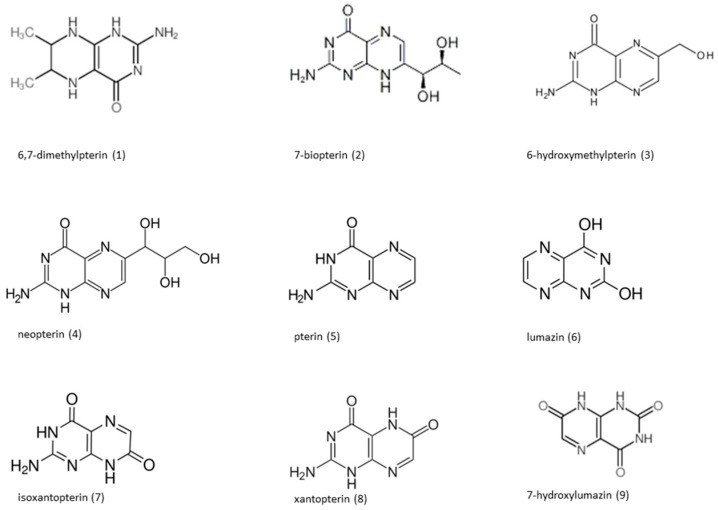
Chemical structures of the selected pterins.

**Figure 2 molecules-24-01166-f002:**
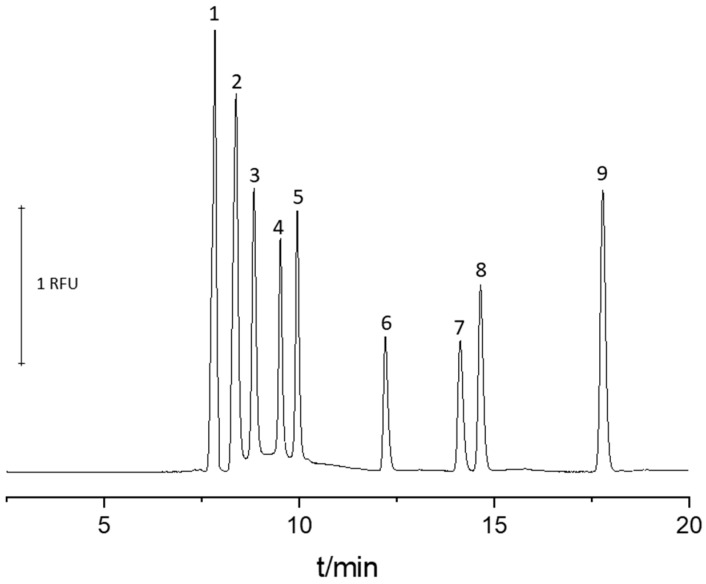
Representative electropherogram obtained from the separation of nine pterins standards under optimized conditions with LED-induced fluorescence detection. Separation was carried out in a fused-silica capillary (35 cm effective length, 56 cm total length, 50 µm internal diameter). The BGE was 100 mM sodium tetraborate, 100 mM TRIS, and 2 mM EDTA sodium salt at pH 9.6. Sample injection was at 50 mbar for 6 s. The analytes concentration was 12 µM. The separation voltage was −17.5 kV. Detection was at λexc/λem = 365/450 nm. Peak identification: 6,7-dimethylpterin (1), 7-biopterin (2), 6-hydroxymethylpterin (3), neopterin (4), pterin (5), lumazin (6), isoxantoptein (7), xantopterin (8), 7-hydroxylumazin (9).

**Figure 3 molecules-24-01166-f003:**
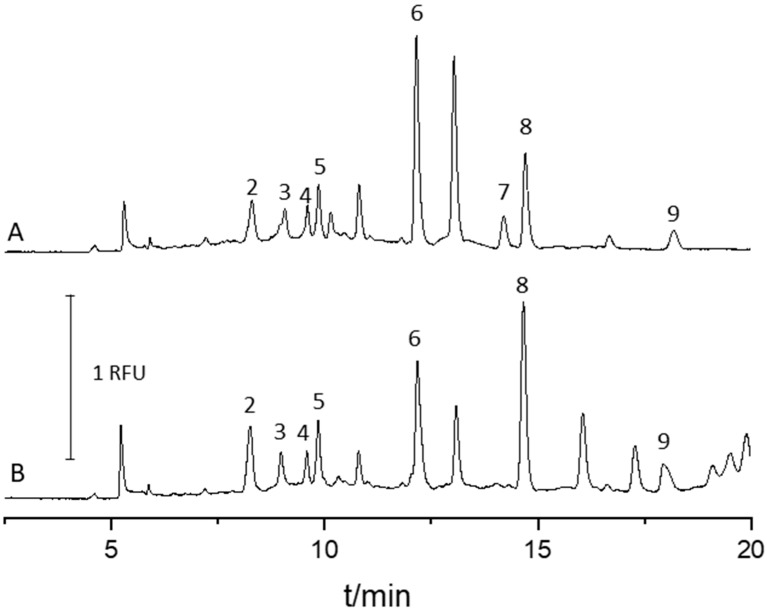
Separation of pterins from a healthy individual (**A**) and a cancer patient (**B**) urine samples. Experimental conditions and peak identity were the same as in Figure 2.

**Table 1 molecules-24-01166-t001:** Analytical figures of merit.

Analyte		Linear Range (µM)	Linear Equation				LOD (µM)	LOQ (µM)
			R^2^	y = ax + b	a ± SD	b ± SD		
1	6,7-dimethylpterin	0.3–15	0.998	y = 6.4243x − 0.0314	0.094	0.002	0.1	0.3
2	7-biopterin	0.3–15	0.996	y = 2.7664x + 0.0009	0.026	0.018	0.1	0.3
3	6-hydroxymethylpterin	0.3–15	0.999	y = 5.7163x − 0.0263	0.036	0.003	0.1	0.3
4	Neopterin	0.3–15	0.999	y = 3.5174x - 0.0099	0.016	0.008	0.1	0.3
5	Pterin	0.3–15	0.999	y = 2.181x + 0.0005	0.017	0.004	0.1	0.3
6	Lumazin	0.3–15	0.999	y = 2.3918x − 0.0048	0.008	0.005	0.1	0.3
7	Isoxantopterin	0.3–15	0.999	y = 1.7642x − 0.0043	0.029	0.002	0.1	0.3
8	Xantopterin	0.3–15	0.999	y = 1.2531x + 0.003	0.012	0.002	0.1	0.3
9	7-hydroxylumazin	0.3–15	0.996	y = 1.4541x + 0.0127	0.020	0.013	0.1	0.3

**Table 2 molecules-24-01166-t002:** Intra-day and inter-day repeatability and precision measured at low, medium, and high concentration level.

Analyte	Concentration		Repeatability		Accuracy (%) (*n* = 5)
	Level	(µM)	Intra-Day %RSD (*n* = 5)	Inter-Day %RSD (*n* = 15)	
6,7-dimethylpterin	LQC	0.75	1.56	4.95	106.20
	MQC	4	2.15	5.33	96.10
	HQC	12	0.80	3.14	91.70
7-biopterin	LQC	0.75	2.53	4.71	110.20
	MQC	4	0.98	0.98	103.60
	HQC	12	2.42	6.59	100.50
6-hydroxymethylpterin	LQC	0.75	1.44	3.92	110.60
	MQC	4	1.54	5.54	98.49
	HQC	12	0.87	7.24	96.28
Neopterin	LQC	0.75	2.61	5.94	106.80
	MQC	4	1.21	6.47	95.66
	HQC	12	1.40	5.48	93.20
Pterin	LQC	0.75	1.94	5.64	102.90
	MQC	4	1.26	6.45	95.70
	HQC	12	0.62	5.34	94.30
Lumazin	LQC	0.75	3.12	6.74	107.40
	MQC	4	1.69	6.85	97.50
	HQC	12	2.10	3.96	92.40
Isoxantopterin	LQC	0.75	3.86	30.45	84.10
	MQC	4	5.22	22.92	79.50
	HQC	12	2.55	22.12	71.20
Xantopterin	LQC	0.75	4.95	5.32	106.70
	MQC	4	2.57	10.57	105.70
	HQC	12	1.63	5.17	94.20
7-hydroxylumazin	LQC	0.75	3.27	7.17	103.30
	MQC	4	3.43	7.36	105.90
	HQC	12	3.53	7.64	106.40

**Table 3 molecules-24-01166-t003:** Samples’ stability in the autosampler.

Analyte		Concentration		Stability%		%RSD
		Level	(µM)	After 3 h	After 6 h	
1	6,7-dimethylpterin	LQC	0.75	94	88	7.0
		HQC	12.00	105	102	5.2
2	7-biopterin	LQC	0.75	89	87	7.1
		HQC	12.00	103	105	4.8
3	6-hydroxymethylpterin	LQC	0.75	89	88	7.1
		HQC	12.00	105	102	5.6
4	Neopterin	LQC	0.75	93	88	7.7
		HQC	12.00	107	102	6.8
5	Pterin	LQC	0.75	95	91	6.1
		HQC	12.00	106	101	6.7
6	Lumazin	LQC	0.75	92	89	6.5
		HQC	12.00	105	101	7.0
7	Isoxantoptein	LQC	0.75	93	91	6.5
		HQC	12.00	106	101	6.9
8	Xantopterin	LQC	0.75	96	93	6.6
		HQC	12.00	105	101	6.6
9	7-hydroxylumazin	LQC	0.75	90	73	14.1

**Table 4 molecules-24-01166-t004:** Samples stability after freezing/thawing cycles.

Analyte		Concentration		Stability%		%RSD
		Level	(µM)	After 24 h	After 48 h	
1	6,7-dimethylpterin	LQC	0.75	94	94	4.1
		HQC	12.00	97	103	3.2
2	7-biopterin	LQC	0.75	84	104	13.1
		HQC	12.00	111	107	5.6
3	6-hydroxymethylpterin	LQC	0.75	91	93	7.6
		HQC	12.00	108	101	4.0
4	Neopterin	LQC	0.75	95	87	9.0
		HQC	12.00	100	98	3.1
5	Pterin	LQC	0.75	98	86	8.9
		HQC	12.00	107	94	6.3
6	Lumazin	LQC	0.75	94	83	11.4
		HQC	12.00	97	88	10.2
7	Isoxantoptein	LQC	0.75	97	62	21.8
		HQC	12.00	91	98	13.4
8	Xantopterin	LQC	0.75	98	92	7.0
		HQC	12.00	101	94	4.4
9	7-hydroxylumazin	LQC	0.75	89	72	17.0
		HQC	12.00	96	92	5.3

**Table 5 molecules-24-01166-t005:** The concentration of pterins measured in urine samples from cancer patients and healthy individuals.

Analyte		Concentration Measured (µM)									
		C1 *	C2	C4	C4	C5	H1 **	H2	H3	H4	H5
1	6,7-dimethylpterin	NA	0.84	1.80	NA	0.36	0.33	0.82	0.59	NA	NA
2	7-biopterin	3.67	1.40	NA	2.43	2.31	3.11	3.04	8.06	0.53	1.73
3	6-hydroxymethylpterin	NA	NA	0.99	1.15	NA	0.31	NA	NA	0.64	1.40
4	Neopterin	2.06	0.72	0.39	1.01	1.05	1.05	0.92	3.37	0.86	0.97
5	Pterin	2.08	0.67	0.49	1.94	1.05	1.10	1.62	4.91	0.57	1.44
6	Lumazin	2.12	NA	0.74	6.43	NA	NA	0.65	1.42	3.43	10.63
8	Xantopterin	0.78	2.48	0.84	8.72	4.94	6.20	5.16	0.80	3.53	4.63
9	7-hydroxylumazin	10.63	0.86	12.29	2.14	3.54	3.64	3.75	3.71	10.5	1.15

* C—urine sample from cancer patient. ** H—urine sample from healthy individual.

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
