# Peer review of "Determination of Urinary Pterins by Capillary Electrophoresis Coupled with LED-Induced Fluorescence Detector"

_molecules, 2019, doi:10.3390/molecules24061166_

Round 1
Reviewer 1 Report
Good beginnig of an interesting job on pterins
Please could mention in you introduction to be complete on CE/UV-LIF: François Couderc, Varravaddheay Ong‐Meang, Véréna Poinsot. Capillary electrophoresis hyphenated with UV‐native‐laser induced fluorescence detection (CE/UV‐native‐LIF) Electrophoresis 2017, 38, 135–149.
Line 118 139 140 143…µM is µmole/L so do not use µM/L, it is wrong.
Line 145-146 “Firstly, it has been found that LIF usually provides two to ten times better sensitivity than LEDIF when the same λexc is used.” It is not always true it depends a lot on the power of the laser, its stability and the power and stability of the LED. So please indicate a reference which agrees to your sentence.
Line 151: I donot understand “The mismatch in λex/λem between LIF and LEDIF detectors could be another factor that affected the sensitivity.”
Line 157: Verify your affirmation, because the ZETALIF is equipped of a PMT not a photodiode.
For me, the main Reason of the difference between LEDIF and LIF is that 325nm is more adapted for excitation of the pterins than the LED 365nm. A 266nm pulsed laser would be still better than 325nm. So please add to your publication (in fig 1) the UV absorbance spectrum of pterins, to show that. And indicate the power of the 365nm LED.
Line 184 "It has been proven that thawing and refreezing of samples can significantly change their composition". Add a reference.
Are you sure that “analytes' concentration” is correct? It should be” the concentration of the analytes”.
Author Response
Reviewer 1
Good beginning of an interesting job on pterins
We thank Reviewer 1 for her/his kind comment.
Please could mention in your introduction to be complete on CE/UV-LIF: François Couderc, Varravaddheay Ong‐Meang, Véréna Poinsot. Capillary electrophoresis hyphenated with UV‐native‐laser induced fluorescence detection (CE/UV‐native‐LIF) Electrophoresis 2017, 38, 135–149.
We thank Reviewer 1 for her/his suggestion. The proposed reference has been added to the revised manuscript. Please see position [34] in references.
Line 118 139 140 143…µM is µmole/L so do not use µM/L, it is wrong.
We thank Reviewer 1 for her/his comment. The error has been corrected in the revised manuscript.
Line 145-146 “Firstly, it has been found that LIF usually provides two to ten times better sensitivity than LEDIF when the same λexc is used.” It is not always true it depends a lot on the power of the laser, its stability, and the power and stability of the LED. So please indicate a reference which agrees to your sentence.
We thank Reviewer 1 for her/his comments. We agree that the difference in sensitivity between LIF and LEDIF detectors might vary significantly and depends on the mentioned factors. According to ZETALIF LED detector specification provided by its supplier (Picometrics) the sensitivity of ZETALIF LEDIF is usually two to five times lower compare to ZETALIF LIF detector. In another study by Sharikova A.V., a comparison between house-built LIF and LEDIF detectors was conducted. The results showed that the fluorescence signal obtained from LED-based system was 1-2 orders of magnitude lower than that from the laser-based system. At the same time, the pulse energy from LED was 2-3 orders of magnitude lower than that from a laser. This also proves the significance of the power of the light source used for excitation.
To clarify the information provided, we rewrote the sentence “Firstly, it has been found that LIF usually provides two to ten times better sensitivity than LEDIF when the same λexc is used” and added the references:
· Sharikova, A.V. UV laser and LED induced fluorescence spectroscopy for detection of trace amounts of organics in drinking water and water sources. Graduate Theses and Dissertations, University of South Florida, Tampa, 2009.
· Zetalif LED. Available online: http://www.adelis-tech.com/product/zetalif-led/(accessed on 15th of March)
The new statement in the revised manuscript is as follows:
‘Secondly, according to the information provided by the manufacturer of the LEDIF detector, its sensitivity in most of the cases is two to five times lower compared to LIF detector when the same λexcis used. Another study, using house-built laser-based and LED-based systems, has shown one to two orders of magnitude lower sensitivity of LED-based system. In both cases, this was a consequence of wider width of wavelengths as well as non-coherent light emitted by the LED.”
Line 151: I do not understand “The mismatch in λex/λem between LIF and LEDIF detectors could be another factor that affected the sensitivity.”
We thank Reviewer 1 for her/his comment. In this sentence, we meant that λex used in the laser-based system (325 nm) induced stronger fluorescence than λex used in the LED-based system (365 nm). The emission signal was collected using band-pass filters in both systems. However, the wavelength range of collected fluorescence was 370-760 nm and 400-539 nm in LED-based system and laser-based system, respectively. Therefore, the differences in both λex and λem used in these two systems could be a reason for different LODs obtained.
To avoid confusion, we deleted “The mismatch in λexc/λem between LIF and LEDIF detectors could be another factor that affected the sensitivity.” in the revised manuscript. However, the discussion about differences between LIF and LEDIF detectors was rewritten according to the previous and next comment of the Reviewer 1.
Line 157: Verify your affirmation, because the ZETALIF is equipped of a PMT, not a photodiode. For me, the main Reason of the difference between LEDIF and LIF is that 325nm is more adapted for excitation of the pterins than the LED 365nm. A 266nm pulsed laser would be still better than 325nm. So please add to your publication (in fig 1) the UV absorbance spectrum of pterins, to show that. And indicate the power of the 365nm LED.
We thank Reviewer 1 for her/his comment. We would like to clarify that we did not say that ZETALIF LED 365 nm was equipped with a photodiode. In the discussion:
“In addition, Ma's group implemented some modifications to their original design to improve the sensitivity. First, at all, they replaced the previously used laser with a more powerful model (35 mW at 325 nm). Another upgrade was replacing 10x microscope lens to 43x microscope lens to reduce background noise arising from non-polarized light scattering off of the outer wall of the capillary. Finally, the photomultiplier tube was used instead of the photodiode detector.”
we were talking about improvements made by Ma’s group on their original house-built LIF detector. According to the previously published papers where fluorescence detection was used for pterins determination, there is a concordance that λexc should be 325 nm. As suggested by Reviewer 1, we added UV absorbance spectra (please see Figure 1 in Supplementary Information) in the revised manuscript. However, the obtained UV absorbance spectra do not clearly indicate that λexc of 325 nm induces stronger fluorescence than 365 nm. The maximum absorbance for most of the analytes was found at around 350 nm which is also consistent with other publications. We believe that the difference in sensitivity between LIF and LEDIF is the resultant of many factors which include light sources used, λexc and λem filters used, and optical system’s design. Since both detectors were built based on different designs, it is difficult to indicate one main reason.
We also think that the results presented in our manuscript might contribute to a broader discussion about differences in sensitivity between LIF and LEDIF detectors. Utilizing the commercially available detectors from Picometrics i.e., LIF at 325, LIF at 355 and LEDIF at 365 nm (where the only variables are a light source or λexc) for pterins determination, could provide the information about the influence of light source and λexc used on the sensitivity.
To address comments of the Reviewer 1 we rewrote the discussion in the revised manuscript. The new statement is as follows:
“There are several possible reasons of such difference in sensitivity between these two systems. Firstly, it has been demonstrated that λexcof 325 nm induces the most intense fluorescence for many pterins. Therefore, Ma’s group used a helium-cadmium laser at 325 nm in their house-built LIF detector. On the other hand, the LEDIF detector was equipped with LED at 365 nm. The obtained UV absorbance spectra (see Figure 1 in Supp. Info.) indicates that the optimum λexc is found at around 350 nm for most of the analytes, which is also consisted with previously published data [36,37]. Secondly, according to the information provided by the manufacturer of the LEDIF detector, its sensitivity in most of the cases is two to five times lower compared to the LIF detector when the same λexc is used. Another study, using house-built laser-based and LED-based systems, has shown one to two orders of magnitude lower sensitivity of LED-based system. In both cases, this was a consequence of wider width of wavelengths as well as non-coherent light emitted by the LED. Finally, the emission signals were collected using band-pass filters in both systems. However, the wavelength range of collected fluorescence was 370-760 nm and 400-539 nm in LED-based system and laser-based system, respectively. Nevertheless, an indication of the main reason for different LODs values obtained by LIF and LEDIF detectors is difficult since both instruments have distinct designs. It should be noticed that no commercially available LIF detector has been used for pterins determination. We believe that utilization of the detectors in which the only variables are a type of the light source or λexc value (e.g., Zetalif LIF 325 nm, Zetalif LIF 355 nm, and Zetalif LED 365 nm from Picometrics), could explain the effect of these variables on the sensitivity in the determination of pterins.”
The power of the LED 365 nm was 100 mW. This information was provided after contacting the manufacturer.
Line 184 "It has been proven that thawing and refreezing of samples can significantly change their composition". Add a reference. Are you sure that “analytes' concentration” is correct? It should be” the concentration of the analytes”.
We thank Reviewer 1 for her/his comments. We agree that the mentioned above statement contains an inaccuracy. After the careful literature review, we found that most of the research articles have shown a possibility rather than a proof of sample degradation during thawing/freezing cycle. To make sure that the provided information is correct we rewrote the mentioned sentence. The new statement in the revised manuscript is as follows: “It has been shown that thawing and refreezing of biological samples might potentially change their composition.” To support this information, we added several references:
· Mitchell, B.L.; Yasui, Y.; Li, C.I.; Fitzpatrick, A.L.; Lampe, P.D. Impact of freeze-thaw cycles and storage time on plasma samples used in mass spectrometry based biomarker discovery projects. Cancer informatics 2005, 1, 98-104.
· Cuhadar, S.; Koseoglu, M.; Atay, A.; Dirican, A. The effect of storage time and freeze-thaw cycles on the stability of serum samples. Biochemia Medica 2013, 23, 70, doi:10.11613/BM.2013.009.
· Hernandes, V.V.; Barbas, C.; Dudzik, D. A review of blood sample handling and pre-processing for metabolomics studies. Electrophoresis 2017, 38, 2232-2241, doi:10.1002/elps.201700086.
· Rotter, M.; Brandmaier, S.; Prehn, C.; Adam, J.; Rabstein, S.; Gawrych, K.; Brüning, T.; Illig, T.; Lickert, H.; Adamski, J., et al. Stability of targeted metabolite profiles of urine samples under different storage conditions. Metabolomics : Official journal of the Metabolomic Society 2017, 13, 4-4, doi:10.1007/s11306-016-1137-z.
· Erben, V.; Bhardwaj, M.; Schrotz-King, P.; Brenner, H. Metabolomics Biomarkers for Detection of Colorectal Neoplasms: A Systematic Review. Cancers2018, 10, doi:10.3390/cancers10080246
We also corrected the phrase “analytes’ concentration” to “the concentration of the analytes” as suggested by Reviewer 1.
Reviewer 2 Report
In submitted publication, authors have focused on quantitative aspects of using light emission diode induced fluorescence detection for electrophoretic determination of pterins in urine. The method has been adopted from published one regarding separation conditions, however the main novelty of the work is in the quantitative parameters, validation and stability study of the compounds. Authors have achieved sensitivity which is lower than using LIF detection.
The manuscript itself is clearly written, data are logically presented and illustrated by the sufficient number of figures and tables. Regarding discussion itself, I have some minor comments and questions:
The determination of LOD and LOQ (presented in Table 1) - were the values calculated from the signal-to-noise ratio or just taken from the lowest sample tested? I wonder if all the analytes have the values the same, as listed in the table.
Table 1 - regression equation - it would be beneficial to provide standard deviations for regression parameters and at least test the significance of intercept. The information about tests of linearity is missing; was it somehow tested, or just deduced that the curves are linear in selected range based on r-square?
Repeatability of determination of isoxanthopterin (misspelled in Table 2) - with respect to the discussion in lines 169-176 and Table 3 - it seems that the stability of the compound is not an issue; can the authors specified mentioned small changes in the conditions affecting the results?
Line 229 - is the reference to Table 1 correct? It does not contain results of determination of pterins.
Author Response
Reviewer 2
In submitted publication, authors have focused on quantitative aspects of using light emission diode induced fluorescence detection for electrophoretic determination of pterins in urine. The method has been adopted from published one regarding separation conditions, however the main novelty of the work is in the quantitative parameters, validation and stability study of the compounds. Authors have achieved sensitivity which is lower than using LIF detection.
The manuscript itself is clearly written, data are logically presented and illustrated by the sufficient number of figures and tables. Regarding discussion itself, I have some minor comments and questions:
We thank Reviewer 2 for her/his kind comments.
The determination of LOD and LOQ (presented in Table 1) - were the values calculated from the signal-to-noise ratio or just taken from the lowest sample tested? I wonder if all the analytes have the values the same, as listed in the table.
We thank Reviewer 2 for her/his comments. To address these questions, the following statement was added to the experimental section of the revised manuscript:
“The LOD and LOQ values were determined by preparing a serial dilution of the stock solution of the analytes (i.e., 5x10-5M, 5x10-6M, 5x10-7M, and 5x10-8M). The concentration of each analyte was the same. Analysis of such prepared dilutions was continued until no analytes were detected. Afterward, the concentration range between the last dilution where analytes were detected and the dilution where analytes were not detected i.e., between 5x10-7M and 5x10-8M was studied in detail. The tested concentration levels were 5x10-7M, 4x10-7M, 3x10-7M, 2x10-7M, 1x10-7M, and 9x10-8M. The LOD and LOQ values were determined when the signal-to-noise ratio was 3 and 10, respectively.”
Table 1 - regression equation - it would be beneficial to provide standard deviations for regression parameters and at least test the significance of intercept. The information about tests of linearity is missing; was it somehow tested, or just deduced that the curves are linear in selected range based on r-square?
We thank Reviewer 2 for her/his comments. The SD values of regressions parameters have been added to Table 1. The linearity of the regression equation was determined based on r-square > 0.995. To clarify this information the following statement was added to the experimental section of the revised manuscript:
“To determine the linear range, ten different concentration levels were used in three replicates each. The lowest concentration was LOQ while the highest one was the maximum concentration that provided linearity with r2 > 0.995.”
In addition, accuracy study, in which three different concentration levels were used, has proven that determined linear range provides good results. The different concentration levels used it that study were low, medium, and high (0.75, 4, and 12 µM) and represented the whole linear range.
Repeatability of determination of isoxanthopterin (misspelled in Table 2) - with respect to the discussion in lines 169-176 and Table 3 - it seems that the stability of the compound is not an issue; can the authors specified mentioned small changes in the conditions affecting the results?
We thank Reviewer 2 for her/his comments. The conducted studies of the stability did not prove the degradation of isoxantopterin under experimental conditions. The precision and accuracy were good for intra-day study whereas, the inter-day study resulted in a significant change. We thought that small changes in the temperature or humidity in the laboratory (not controlled) could possibly influence the results. On the other hand, the rest of the analytes seemed not to be affected by such changes. Based on the experiments that we performed be are unable to prove any of those hypotheses. We think that more detailed studies are required to explain the poor inter-day repeatability of isoxantopterin. To address this issue, we added the following statement to the revised manuscript:
“On the other hand, the poor inter-day repeatability of isoxantopterin cannot be explained based on the stability tests conducted. Therefore, further stability studies, including a longer period of time, the influence of temperature and humidity, are required to address this issue.”
Line 229 - is the reference to Table 1 correct? It does not contain results of determination of pterins.
We thank Reviewer 2 for her/his comment. The reference in line 229 should be to Table 5. It has been corrected in the revised manuscript.
Round 2
Reviewer 1 Report
good job